# Whole genome sequencing of *Yersinia pestis* isolates from Central Asian natural plague foci revealed the role of adaptation to different hosts and environmental conditions in shaping specific genotypes

Aigul A. Abdirassilova[1]*, Duman T. Yessimseit[1], Altynai K. Kassenova[1], Beck Z. Abdeliyev[1], Zauresh B. Zhumadilova[2], Gulnara Zh. Tokmurziyeva[2], Galina G. Kovaleva[2], Ziyat Zh. Abdel[3], Tatiyana V. Meka-Mechenko[3], Saule K. Umarova[2], Elmira Zh. Begimbayeva[4], Sanzhar D. Agzam[4], Vladimir L. Motin[5], Oleg N. Reva[6], Altyn K. Rysbekova[1]*

1 National Scientific Center of Especially Dangerous Infections named after Masgut Aikimbayev, Laboratory of Molecular-Genetic Studies, Almaty, Kazakhstan, 2 National Scientific Center of Especially Dangerous Infections named after Masgut Aikimbayev, Almaty, Kazakhstan, 3 National Scientific Center of Especially Dangerous Infections named after Masgut Aikimbayev, Laboratory of Plague Microbiology, Almaty, Kazakhstan, 4 National Scientific Center of Especially Dangerous Infections named after Masgut Aikimbayev, National Collection of Microorganisms, Almaty, Kazakhstan, 5 The University of Texas Medical Branch, Dep. of Pathology, Galveston, Texas, United States of America, 6 Centre for Bioinformatics and Computational Biology, Department of Biochemistry, Genetics and Microbiology (BMG), University of Pretoria, Pretoria, South Africa

* aigul.abdirassilova@mail.ru (AAA) rysbekova23@mail.ru (AKR)

## Abstract

The genetic diversity and biovar classification of *Yersinia* isolates from Central Asia were investigated using whole-genome sequencing. In total, 98 isolates from natural plague foci were sequenced using the MiSeq platform. Computational pipelines were developed for accurate assembly of *Y. pestis* replicons, including small cryptic plasmids, and for identifying genetic polymorphisms. A panel of 99 diagnostic polymorphisms was established, enabling the distinction of dominant Medievalis isolates derived from desert and upland regions. Evidence of convergent evolution was observed in polymorphic allele distributions across genetically distinct *Y. pestis* biovars, *Y. pseudotuberculosis*, and other *Y. pestis* strains, likely driven by adaptation to similar environmental conditions. Genetic polymorphisms in the *napA*, *araC*, *ssuA*, and *rhaS* genes, along with transposon and CRISPR-Cas insertion patterns, were confirmed as suitable tools for identifying *Y. pestis* biovars, although their homoplasy suggests limited utility for phylogenetic inference. Notably, a novel cryptic plasmid, pCKF, previously associated with the strain of the population 2.MED0 from the Central-Caucasus high-altitude autonomous plague focus, was detected in a genetically distinct isolate of 2.MED1 population from the Ural-Embi region, indicating potential plasmid transfer across the 2.MED lineage. These findings emphasize the

**Data availability statement:** All sequenced genomes are available from NCBI from BioProject PRJNA1249055 (https://www.ncbi.nlm.nih.gov/bioproject/1249055). All program codes created for this project are available at Zenodo database: Illamina – Python3 pipeline for Linux/Ubuntu to perform bacterial genome assembly from Illumina paired-end reads. Available at https://zenodo.org/records/15227656. doi: 10.5281zenodo.15227656; YPPA – Python3 pipeline for Linux/Ubuntu to perform assembly of *Yersinia pestis* plasmids using Illumina paired-end reads. Available at https://zenodo.org/records/15227829. doi: 10.5281/zenodo.15227829 VARCALL - Python3 pipeline for Windows/Linux/Ubuntu to predict patterns of polymorphic sites in *Yersinia pestis* genomes. Available at https://zenodo.org/records/15228498. doi: 10.5281/zenodo.15228498.

**Funding:** This work was supported by the Minister of Science and Higher Education of the Republic of Kazakhstan; Grant AP19680079 to AAA, "Study of molecular genetic features and variability of plague and tularemia strains in epidemiological surveillance of zoonoses" to AAA. The funder had no role in study design, data collection and analysis, decision to publish, or preparation of the manuscript.

**Competing interests:** The authors have declared that no competing interests exist.

need for ongoing genomic surveillance to monitor the spread of virulence-associated genetic elements and to improve our understanding of *Y. pestis* evolution and ecology.

## Author summary

The central hypothesis of this study was that genetic variation in *Y. pestis* may correlate with host or environmental origin potentially obscuring clonal ancestry, thereby facilitating pathogen surveillance and offering new insights into its evolutionary dynamics. This study investigates the genetic diversity of *Yersinia pestis* isolates from Central Asian natural plague foci. A panel of diagnostic genetic polymorphisms was developed to differentiate between sublineages, revealing a clear division of the dominant Medievalis (2.MED) biovar into desert and upland branches. These populations appear to reflect ecological adaptation rather than strict phylogenetic lineage, suggesting that convergent evolution plays a significant role in shaping *Y. pestis* diversity. A novel cryptic plasmid, pCKF, previously found only in Caucasus isolates 2.MED0, was detected in an unrelated desert strain from Kazakhstan, indicating possible horizontal gene transfer. Additionally, analysis of mobile genetic elements, including transposons and CRISPR-Cas inserts, provided further resolution of strain relationships and evolutionary dynamics. The developed diagnostic panel, transposon distribution patterns and replicon assembly pipeline developed for this project offer valuable tools for plague surveillance and monitoring in nature. The study underscores the importance of ecological and genomic monitoring to understand the evolution, adaptation, and potential spread of *Y. pestis* in endemic regions.

## Introduction

Plague outbreaks inspire fear in people due to the historical memory of three devastating epidemics: the Justinian Plague (541–542 CE), which killed an estimated 25–50 million people (around 40% of the population); the Black Death (1347–1351 CE), which caused 75–200 million deaths and wiped out approximately 30–60% of Europe's population; and the Third plague pandemic (1855–1959), which resulted in 12 million deaths in India and China alone, with outbreaks in other parts of the world [1–4]. Based on a correlation between the current geographical sources of the isolates it was suggested that the causative agents of these pandemics were three biovars of *Yersinia pestis* – Antiqua (ANT), Medievalis (MED), and Orientalis (ORI), respectively [5]. This biovar classification was based on biochemical properties of the isolates, and lately on several genetic differences, allowing their identification in exhumed victims of the pandemics [6]. In the latter study, the term "subspecies" of *Y. pestis* was used, which is arguably an overestimate given the extreme genetic homogeneity of the pathogen [7]. Nevertheless, the sequencing of ancient genomes

revealed that both the Justinian and Black Death pandemics ware caused by the *Y. pestis* strains of different phylogenic branches [8–10]. The existence of strains with limited virulence isolated mostly in high-mountain regions were classified originally into Microtus biovar [11] and then as non-main *Y. pestis* subspecies often called "pestoides" [12].

*Y. pestis* virulence depends heavily on three plasmids: the large pMT1, the medium pCD1, and the small pPCP1. The pMT1 plasmid encodes the F1 capsule and Ymt toxin, which prevent phagocytosis and enable bacterial survival in the flea, respectively [13]. The pCD1 plasmid harbors the Type III Secretion System, including *lcrV* and *yopS* genes, which actively suppress host immunity [13,14]. The pPCP1 plasmid encodes Pla protease that facilitates tissue invasion and is essential for pneumonic infection [15]. Taken together, the presence of these three plasmids is a stable hallmark of most *Y. pestis* isolates, underpinning their high virulence and efficient transmission cycles [13]. However, a new the shortest plasmid pCKF of 5.4 kbp has been discovered recently in *Y. pestis* isolated from Central Caucuses [16,17]. The role of this plasmid in pathogen virulence remains unclear.

In addition to the plasmids, transposable elements are abundant in *Y. pestis* genomes and play a significant role in genetic mobility, genome plasticity, and the evolution of this pathogen [18]. They facilitate the mobilization of adjacent genes, contributing to the spread of virulence factors. For example, IS100 is associated with the plasminogen activator gene (*pla*), a key virulence factor in *Y. pestis* [13,19].

There are many regions globally, primarily between latitudes 55° North and 40° South, where *Y. pestis* persists as a zoonotic disease in wild rodent populations and their fleas, sporadically affecting people living in these areas, known as natural plague foci [2–4,20–23]. For instance, the natural endemic range of plague in Kazakhstan covers an area of approximately 1,117,000 km², constituting about 41% of the country's territory. Within this expanse, there are six natural and 15 autonomous plague foci, encompassing over 90 landscape-epizootological districts. These foci are integral to the Central Asian desert natural plague focus, which is among the most extensive natural plague centers globally. The primary reservoirs of *Y. pestis* in these regions include gerbils (*Rhombomys opimus*), ground squirrels (*Spermophilus* spp.), and marmots (*Marmota* spp.), with fleas from the genera *Xenopsylla* and *Nosopsyllus* serving as the main vectors. The extensive distribution of these natural plague foci underscores the importance of continuous surveillance and targeted control measures to mitigate the risk of human plague outbreaks in Kazakhstan. These measures include regular sample collection for pathogen monitoring by specialized anti-plague stations, vaccination of at-risk populations and camels, disinfection and deratization of villages and livestock shelters, and public education efforts [23].

There is a growing concern that, due to intensive agricultural and industrial activity in these areas, booming tourism, human migration, and climate change, plague outbreaks may spiral out of control and affect many people worldwide [3,24–26]. Plague outbreaks can occur even in industrially advanced countries, as evidenced by the outbreak in Colorado (USA) during June-July 2014 [27], when four separate human plague infections were diagnosed. These included the owners of an infected dog and veterinary workers, all of whom became infected after exposure to the sick animal.

Several mutations in the evolutionary pathway from *Yersinia pseudotuberculosis* to the highly pathogenic *Y. pestis* have been reported. For instance, a frameshift mutation in the Rcs signaling pathway [28] and the activation of the *hms* (hemin storage) gene cluster through disrupted cyclic di-GMP regulation [29] caused biofilm-induced blockage in the flea foregut. This blockage led to repeated biting behavior by infected fleas as they attempted to feed and clear the obstruction. Such behavior dramatically increased the transmission efficiency of *Y. pestis* to mammalian hosts, including humans. Also, the *ureD* mutation resulted in silencing urease likely provided a strong positive selection for the increase transmission potential by fleas [30].

Species identification of *Y. pestis* and biovar identification have traditionally been performed using phenotypic tests, such as carbohydrate fermentation and the ability to denitrify. Loss of glycerol fermentation and nitrate reduction activities has occurred in modern *Y. pestis* biovars, such as ORI and MED, respectively. However, the ANT biovar retains both these activities [11].

Denitrifying activity, the ability to reduce nitrates to nitrites, is one of the key diagnostic tests used to differentiate *Y. pestis* biovars. For instance, the *Y. pestis* biovar MED lacks this ability (nit−), while the ANT and ORI biovars exhibit it (nit+). The denitrification reaction involves the reduction of $NO_3^-$ anions, leading to the formation of various products, which are detected using Griess reagent by the appearance of a crimson coloration in the medium [5,31]. MED strains are nit−due to a truncation of the periplasmic reductase *napA* [11,32,33]. Remarkably, unrelated strains of the non-main *Y. pestis* subspecies Altaica and Hissarica developed a similar nit− phenotype due to an alternative mutation in the nitrate-binding domain of the ABC transporter protein SsuA [34]. The convergent development of similar phenotypic traits suggests the evolutionary significance of these mutations. It could be that the strains with compromised metabolism are more virulent for the particular host; however, the correlation between *Y. pestis* pathogenicity and the loss of nitrate reduction ability remains an area of ongoing research.

Carbohydrate fermentation tests are used for intra- and interspecies differentiation of the plague microbe, particularly in the identification of *Y. pestis* and *Y. pseudotuberculosis* pathogens [5,31]. For instance, the glycerol fermentation test is one of the key diagnostic markers differentiating *Y. pestis* biovar ORI, which is incapable of fermenting glycerol (Gly−), from biovars ANT and MED, which can ferment it (Gly+). Glycerol fermentation was lost in *Y. pestis* strains of the ORI biovar due to deletions in the *glpD* and *glpK* genes, encoding glycerol-3-phosphate dehydrogenase and aerobic glycerol kinase, respectively [35,36].

Rhamnose fermentation is an intra-species differentiation test for the plague microbe and is considered specific for identifying a non-main biovars of *Y. pestis*. The emergence of Rha+ strains is likely an adaptation of the pathogen to certain mammalian hosts (such as voles and the Mongolian pika) and their ectoparasites. The Rha+ trait correlates with selective virulence for white mice and certain wild animal species, sensitivity to pesticin 1, and the requirement for additional growth factors [37]

The vast majority of *Y. pestis* strains ferment arabinose (Ara+). However, this trait is absent in Rha+ strains from strains of the non-main biovars from the Altai Mountains and the Gissar Ridge. In contrast, Rha+ isolates of non-main biovars from the Trans-Caucasus Highlands and certain regions of Mongolia ferment arabinose similarly to Rha− strains of the main biovars of *Y. pestis* [37].

While biochemical tests and PCR-based VNTR/MLVA typing are widely used for the characterization of *Y. pestis* isolates [38,39], these approaches are sometimes inconclusive for atypical strains and do not provide sufficient resolution for monitoring individual clonal lineages of the pathogen in nature.

Advances in sequencing technologies have enabled the massive sequencing of *Y. pestis* isolates for comprehensive profiling of their genetic polymorphism. This approach could potentially allow the timely detection of marker mutations and other genetic aberrations in zoonotic populations of the pathogen, which may lead to new disease outbreaks. Such surveillance should also include monitoring mobile genetic elements. For instance, filamentous bacteriophages, such as YpfΦ, have been implicated in the virulence and evolution of pathogenic bacteria, including the ORI biovar of *Y. pestis* [40,41]. Conventionally, the number of CRISPR-Cas inserts and the lengths of their repeated regions may influence the likelihood of the emergence of new highly virulent variants of the pathogen by controlling prophage acquisitions. Of particular concern is the recent reporting of a new cryptic plasmid, pCKF, isolated from *Y. pestis* of the Central-Caucasus plague focus [42,17].

The focus of the current study was the assessment of genetic variability in *Y. pestis* strains collected from the vast area of Kazakhstan and border regions of Kyrgyzstan, encompassing various autonomous plague foci and landscapes. The aim was to identify novel marker mutations and to associate genetic polymorphisms with either ancestral inheritance or convergent evolution driven by adaptations to similar environmental conditions, mammalian hosts, or vector insects. The central hypothesis was that genetic variation in *Y. pestis* may correlate with host or environmental origin potentially obscuring clonal ancestry, thereby facilitating pathogen surveillance and offering new insights into its evolutionary dynamics.

## Materials and methods

### Yersinia strains used in this study

*Yersinia* strains used in this study were selected from the National Collection of MicroOrganisms (NCMO) at the National Scientific Center of Especially Dangerous Infections named after Masgut Aikimbayev (NSCEDI, https://nscedi.kz/en/) in Almaty, Kazakhstan. All work involving *Yersinia pestis* was conducted in full accordance with national biosafety regulations and international standards for handling BSL-3 agents. Routine surveillance, strain maintenance, and research activities are carried out under institutional biosafety protocols approved by the Biosafety and Ethics Committee of the Institute. The use of *Y. pestis* strains from the NCMO for this study did not involve work with live animals or human subjects and was performed exclusively on archived bacterial cultures in compliance with national and institutional regulations.

Strains were isolated from different regions during field expeditions conducted for sample collection from natural plague foci in Central Asia, spanning a long period beginning in 1961 (the earliest strain) and continuing through to recent years, with the latest strain collected in 2022 (see S1 Table, where strains are ordered by the plague foci of their isolation). The selected strains include both typical and several atypical isolates representing various desert and mountain plague foci of Central Asia. Precise coordinates of the isolation sites were recorded and plotted on the map shown in Fig 1 using Arc-Map 10.8 (https://desktop.arcgis.com/en/arcmap/), installed at the NCMO. The administrative boundaries on the map were obtained from the GADM database of Global Administrative Areas (version 2.8) accessible at https://gadm.org.

As control, the vaccine strain *Y. pestis* EV76, used for the production of the live attenuated plague vaccine EV and characterized by typical cultural, morphological, and biochemical properties, was utilized, along with *Y. pseudotuberculosis* O1 with the NCMO registration number 2841.

Screening of the phenotypic properties and characteristics of the studied *Y. pestis* strains was conducted using traditional microbiological methods, including light and fluorescence microscopy (MAGUS Lum 400L microscope), assessment of growth on nutrient media, enzymatic and denitrifying activity, and sensitivity to specific bacteriophages.

### Nutrient media used for *Y. pestis* cultivation and diagnostics

The strains were cultivated on Hottinger liquid and agarous media for 48 hours at 28°C. Their cultural and morphological properties were studied following standard laboratory diagnostic methods described previously [43]. Colony formation was monitored using light microscopy after 12, 24, and 48 hours of cultivation. Another diagnostic medium used in this study was Hottinger agar with 1% blood hemolysate.

### Diagnostic bacteriological tests

**Fermentation and denitrification.** Tests used in this study were described in publication by Eroshenko et al. (2023) [44].

The tested strains were inoculated into 5 ml of Hiss medium (1% peptone water, 1% Andrade indicator, pH 7.2) containing 1% arabinose, rhamnose, or glycerol. Positive fermentation was determined by a color change in the medium (from yellow to pink) after 48 hours of incubation at 28°C.

To determine the denitrification activity, a loopful of a 24–48-hour agar culture of *Y. pestis* was inoculated into 1 ml of Hottinger broth (pH 7.2) supplemented with 0.1% potassium nitrate ($KNO_3$). The cultures were incubated at 28°C for 72 hours, after which 0.5 ml of Griess reagent was added. If denitrifying activity was present, the medium turned crimson upon the addition of the reagent.

**Lysis by *Y. pestis*-specific and *Y. pseudotuberculosis*-specific phages.** In this study, two phages widely used for the diagnosis of *Yersinia* were employed: a broad-spectrum Diagnostic Pseudotuberculosis Bacteriophage (Russia, Saratov, series No. 20, manufactured in 04.2022, valid until 04.2025); and the Pokrovskaya *Y. pestis*-specific bacteriophage (Diagnostic Plague Bacteriophage Pokrovskaya, Russia, Saratov, series No. 21, manufactured in 12.2022,

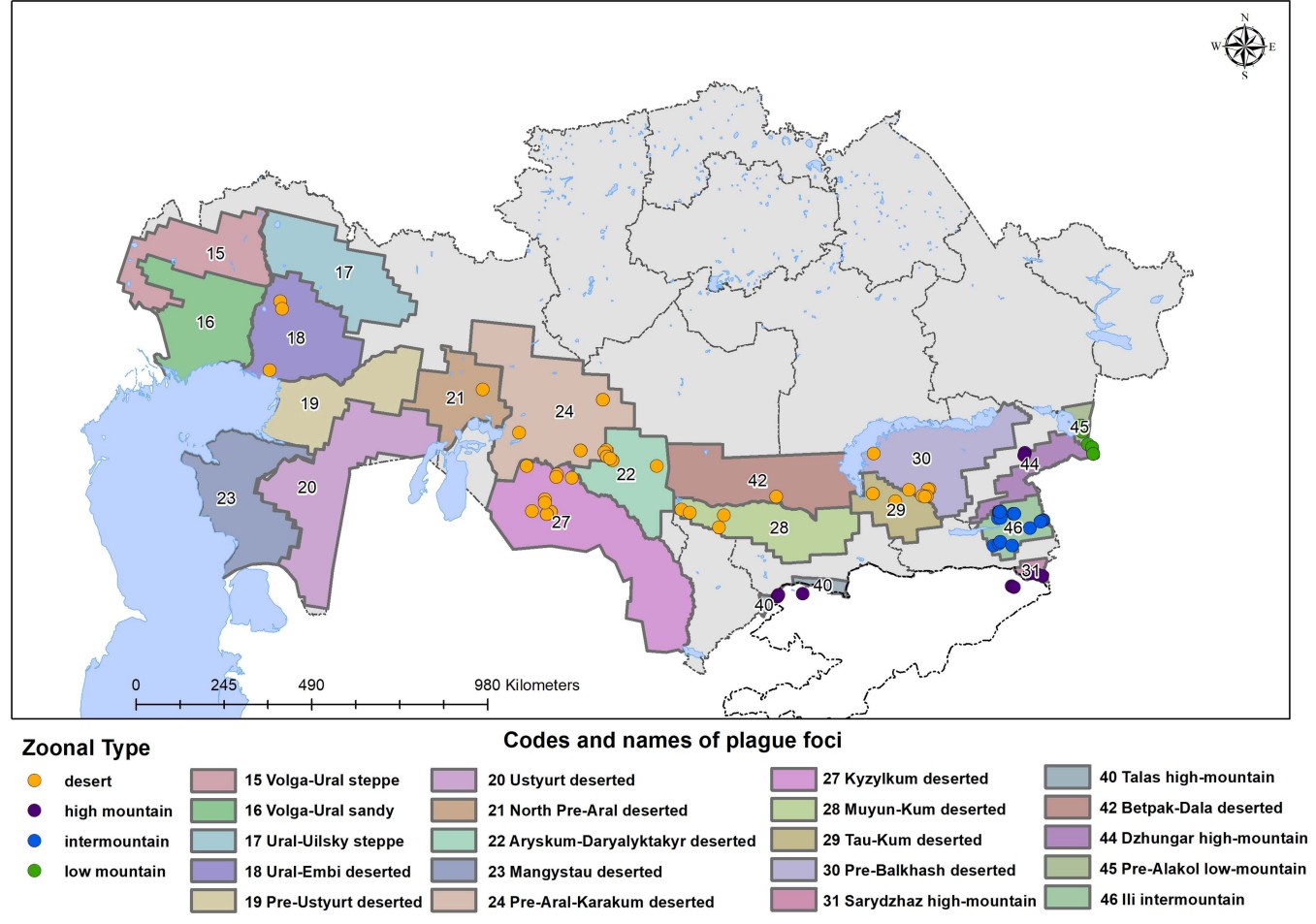

**Fig 1. Regions of sampling the *Yersinia* strains used in this study plotted using the program ArcMap.** The administrative boundaries on the map were obtained from the GADM database of Global Administrative Areas (version 2.8) accessible at https://gadm.org.

valid until 12.2025) [45]. The sensitivity of *Y. pestis* isolates to these phages was tested using the streak-and-spot method. Cultures of selected *Y. pestis* strains, grown at 28°C for 48 hours, were streaked onto Hottinger agar plates, and the bacteriophages were applied as spots near the streaks of inoculated bacteria [46].

The agar plates with applied bacteriophages were dried and incubated in a thermostat at 28°C for 15–18 hours. If the tested culture belongs to the species *Y. pestis*, no growth is observed at the spot of bacteriophage application. *Y. pseudo-tuberculosis* bacteria are lysed only by the pseudotuberculosis bacteriophage and are not susceptible to *Y. pestis*-specific phages.

## Polymerase Chain Reaction (PCR) analysis

A set of primers shown in Table 1 was recommended for *Y. pestis* biovar differentiation [47].

The PCR reaction was carried out in a 25 µl reaction mixture containing 5 µl of 5X Genta PCR master mix, which includes TaqF DNA polymerase, dNTPs, $Mg^{2+}$, and buffer, as well as 1 µl of each primer (forward and reverse) at a concentration of 20 pmol/µl, 5 µl of template DNA, and 8 µl of nuclease-free water to achieve the required volume. The amplification process was conducted on a Rotor-Gene Q thermocycler (QIAGEN, Germany) and started with an initial

**Table 1. Diagnostic primers used in this study.**

| Primer sets or target genes | Primer direction | Sequences | Expected PCR products |
|---|---|---|---|
| med24 | F-primer | GTATTTTGTGTCACCCC | 198 bp/ 222 bp − MED/ANT |
| | R-primer | AATGAGACACCGCCAGT | |
| 2.ANT/2.MED | F-primer | AAGACCTTCGCCACCAGA | 397 bp/ 467 bp − MED/ANT |
| | R-primer | CCAGGATTCGCCGATTCA | |
| *glpD* | F-primer | GGCTAGCCGCCTCAACAAAAACAT | 415 bp/ 508 bp − ORI / MED & ANT |
| | R-primer | GGTCATACAAGAACAAGCCGGTGC | |
| pCKF | F-primer | AAGACCTTCGCCACCAGA | 420 bp if the plasmid is present |
| | R-primer | CCAGGATTCGCCGATTCA | |

denaturation at 95°C for 5 minutes, followed by 35 cycles, each consisting of denaturation at 95°C for 35 seconds, annealing at 60°C for 35 seconds, and elongation at 72°C for 35 seconds. The final step involved a final elongation at 72°C for 10 minutes. The amplified PCR products were analyzed using electrophoresis in a 1.5% agarose gel stained with ethidium bromide for DNA visualization. The electrophoresis was performed using equipment from Amersham Pharmacia Biotech (Sweden). The sizes of the amplicons were determined using the Thermo Scientific GeneRuler 100 bp DNA ladder. Visualization of the amplified products was conducted in the UVP Mini Darkroom system (UVP, USA) using UV light to observe the fluorescence of the bands.

## DNA extraction and sequencing

The cell suspensions of the studied *Y. pestis* strains were subjected to heat treatment at 100°C for 10 minutes. The suspensions were then centrifuged for 2 minutes at 12,000 rpm using a Microlite RF centrifuge (Thermo Electric Corporation, USA, serial number 35830164). The resulting supernatant was used for DNA extraction.

DNA extraction was performed using the QIAamp DNA Mini Kit (QIAGEN, Germany, cat. no. 51304, batch no. 172042359) according to the manufacturer's protocol. The quality of the extracted DNA was assessed using a NanoDrop 1000 spectrophotometer (Thermo Fisher Scientific, USA, serial number B651) by measuring the 260/280 nm ratio and by electrophoresis in a 1% agarose gel stained with ethidium bromide. All manipulations were performed under sterile conditions using disposable filter tips, gloves, and sterile tubes to minimize the risk of sample contamination. DNA sequencing was performed on the MiSeq platform (Illumina, USA) following the manufacturer's instructions. The MiSeq Reagent Kit v3 was used for library preparation. Samples were diluted to a total concentration of 20 ng in 10 µl and used for fragmentation. As a result, PCR products were enzymatically fragmented to a target insert size of 600 bp and ligated with index adapters.

Next, DNA libraries were amplified for 7 PCR cycles using i5 and i7 indices from the Nextera DNA CD Indexes, following the Nextera DNA Flex Library Prep Kit protocol. After amplification, DNA libraries were purified using AMPure XP magnetic beads (Beckman Coulter, USA) by adding them to the reaction mixture at a 1.8:1 ratio, incubating at room temperature for 5 minutes, and performing two washes with 80% ethanol. The pellets were resuspended in 25 µl of TE buffer and incubated for 2 minutes at room temperature.

DNA libraries were diluted using the standard normalization method to a final concentration of 4 nM, assuming an insert size of 600 bp, following the MiSeq System Denature and Dilute Libraries Guide. The samples were then pooled (combined into a single tube) with 3 µl of each library. The pooled DNA library was diluted to 15 pM for sequencing. The loading libraries contained phiX Control v3 at a final concentration of 1%, as recommended by Illumina. The diluted and denatured DNA library was added to the reagent cartridge in a volume of 600 µl. The preparation and sequencing run on the MiSeq system were performed according to the MiSeq System Guide.

## Read quality control and assembly

The genome assembly pipelines for *Y. pestis* chromosome and plasmid assembly, which combine reference-based and de novo assembly steps, as shown in S1 Fig, are implemented in freely available Python3 scripts [48,49]. The pipeline was run on the Computer Server at the Centre for Bioinformatics and Computational Biology (CBCB) at the University of Pretoria (http://wiki.bi.up.ac.za/wiki/index.php/Hardware_resources). An additional Python3 pipeline was developed for calling genetic polymorphisms in *Y. pestis* genomes [50] that will be explain in detail in the Results.

Archived *.fastq.gz files were unzipped and quality-controlled using Trimmomatic. The trimmed DNA reads were used for de novo assembly with SPAdes v3.15.0 and for mapping against reference sequences using Bowtie2. The genome of *Y. pestis* strain SCPM-O-B-6899 (CP045145) was used as the reference. This strain was selected because it is equidistant from all current isolates and possesses the rare plasmid pCKF [16,17]. Consensus sequences were generated from the aligned reads using BCFtools.

Contigs assembled by SPAdes were filtered to remove small sequences (<5,000 bp). In the next step, consensus sequences assembled from the reads mapped against the reference sequences and the de novo assembled contigs were merged using RagTag v2.1.0. This program first creates scaffolds by aligning the contigs to the consensus sequence, and then patches missing parts of the consensus sequence from the de novo assembled contigs. Bowtie2 v2.4.1 mapping of the original DNA reads against the patched consensus sequences is used to remove false patches and smooth the borders between patches and the consensus sequence. A final consensus sequence is generated using BCFtools v1.18. The resulting genomic consensus sequences were annotated with Prokka v1.13.3 and deposited at the NCBI under BioProject PRJNA1249055. The respective BioSample accession numbers are listed in S1 Table.

## Statistical validation of clade segregation by genetic polymorphisms

Clade segregation accuracy was assessed using the random forest classification algorithm, implemented via the *RandomForestClassifier* and *model_selection.cross_val_score* functions from the Python library *sklearn* (version 1.7.0). To evaluate whether the observed distribution of strains among clusters based on genetic polymorphism analysis significantly differs from a random distribution, a bootstrap analysis with 1,000 iterations was performed. In each iteration, the strains were randomly assigned to clusters, and classification accuracy scores were calculated. The likelihood that the accuracy from a random assignment equals or exceeds the actual score was estimated as a p-value using the *mean* function from the NumPy library (version 2.2.4).

Additionally, PERMANOVA and Mantel tests for assessing the statistical significance of clade segregation were applied using the implementations provided in the *stats.distance* module of the *scikit-bio* package (v.0.7.0). An in-house Python 3 pipeline, *yp_classifier*, was developed for this project to perform all of the aforementioned statistical tests. The pipeline is available at https://zenodo.org/records/16875681.

Confidence interval (CI) of cross-population introgressions was calculated using the Wilson score interval [51], which is applicable for analyzing genotype-ecotope associations in microbiology [52] (Eq. 1).

$$CI = \frac{p + pz^2 \mp z\sqrt{\frac{p(1-5)}{n} + \frac{z^2}{4n^2}}}{1 + \frac{z^2}{n}}$$

(1)

where $p$ is the observed proportion of introgressions; $n$ is sample size; $z = 1.96$ for a 95% CI.

## Results

### Strain isolation and bacteriological testing

In total, 98 *Yersinia* strains were selected from the National Collection of MicroOrganisms (NCMO) at the National Scientific Center of Especially Dangerous Infections named after MasgutAikimbayev (NSCEDI), covering 12 autonomous

plague foci, including desert and upland landscapes, as described in a previous publication [23]. The locations of isolation points are shown in Fig 1 and detailed in S1 Table.

Y. pestis isolates formed "ground glass" colonies after 12 hours of cultivation on Hottinger agar at 28°C (S2A Fig). These colonies transitioned into semi-transparent colonies with scalloped edges after 24 hours of cultivation (S2B Fig). By 48 hours, typical greyish-white colonies developed, characterized by a convex, darker, finely granular or rough centre and a flat, undulating edge resembling lace, also known as R-type colonies (S2C Fig). On the Hottinger agar with 1% blood hemolysate, Y. pestis strains produced granular colonies with a very small lace-like zone shown in S2D Fig.

The results of laboratory testing for the selected isolates are shown in S2 Table. All selected strains were susceptible to lysis by anti-Y. pestis phage and the anti-Y. pseudotuberculosis phage, except for the strain 53_YP3_IM, which was resistant to the Y. pestis-specific Pokrovskaya phage. This strain, along with three other strains isolated from the high-mountain Talas plague focus (35_YP14_TLH, 36_YP27_TLH, and 37_YP35_TLH), were able to ferment rhamnose, whereas all other isolates were rhamnose-negative. All tested strains fermented glycerine, and the majority fermented arabinose, except for the three Talas strains mentioned above. The strains under study varied in their denitrification activity, with six strains being nit+ and all others nit −.

PCR amplification targeting the variable rpoB region did not reveal the deletion characteristic of the ORI biovar in any of the isolates, and primers targeting the variable region within glpD showed no polymorphism between the studied isolates. The amplification products obtained with MED-specific primer sets (med24 and ANT/MED) identified the majority of the strains as belonging to the MED biovar. However, the strains 26_YP30_SZ, 27_YP31_SZ, 28_YP29_SZ, 31_YP22_SZ, 38 YP25_SZ and 34_YP28_SZ (all nit+), four Talas isolates (including 39_YP13_TLH), and strains 42_YP10_UE and 53_YP3_IM produced longer PCR amplicons with these primers distinguished from the amplicon's characteristic for MED isolates.

### Y. pestis biovar identification and typing using whole genome sequences

Whole-genome sequences were assembled using the in-house ILLAMINA pipeline and YPPA pipelines for assembly of chromosomes and plasmids, respectively [48,49]. Each replicon (chromosome or plasmid) was represented by a single complete sequence without gaps. N's in the sequences indicated nucleotide positions of known length that were characterized by low quality or ambiguous base calls. The GC content of the assembled chromosomes ranged from 47.5% to 47.6%. Genome assembly statistics, including assembly length, GC%, and mean sequencing depth (coverage) calculated for each replicon, are shown in S3 Table. The availability of whole-genome sequences for the isolates enabled computer-based verification of the laboratory results of biovar identification and VNTR typing.

Y. pestis strains of the main 'modern' biovars can be distinguished from Y. pseudotuberculosis and the non-main 'ancestral' Y. pestis Microtus biovar by comparing sequences of the acetolactate synthase small subunit gene, ilvN [53,54]. Main Y. pestis strains have a deletion at the 3' end of the gene, making it 45 bp shorter. The analysis of assembled genomes revealed that the majority of the isolates belonged to the main biovars except for the strains 57_YP22_IM, 53_ YP3_IM from Ili intermountain focus, and 36_YP27_TLH, 39_YP13_TLH, 37_YP35_TLH and 35_YP14_TLH from Talas high mountains. The ilvN sequences in 57_YP22_IM и 53_YP3_IM were identical to those in the reference Y. pseudotuberculosis isolate. However, in the four strains isolated from the Talas plague focus in Kyrgyzstan, a non-sense A→C substitution at the 51st nucleotide location was detected.

The biovar ORI can be distinguished from other biovars by a 93 bp deletion in aerobic glycerol-3-phosphate dehydrogenase glpD [11,35]. None of the isolates had this deletion that was in good correspondence with the actual glycerine fermentation test.

An insertion of G at the 773rd position in the arabinose operon regulator araC, leading to gene truncation due to the appearance of a premature stop codon, is characteristic of strains of the non-main 0.PE4 lineages isolated in the Altay Mountains and the Gissar Range in Tajikistan [55]. Consequently, truncated araC was found in our four non-main biovar strains from Talas. According to Kislichkina et al. (2015), the Gissar isolates could be distinguished from the sister Altay

strains by a G→A substitution at the 482$^{nd}$ nucleotide position in the rhamnose operon transcriptional activator *rhaS* [56]. This substitution has not been observed in the analyzed strains.

Another polymorphism in *rhaS* has been reported at the 671$^{st}$ position [33,57]. All strains of the modern *Y. pestis* biovars have A at this position, while *Y. pseudotuberculosis* and ancestral *Y. pestis* have G. A guanine residue was found in six analyzed strains: the reference *Y. pseudotuberculosis* IP 32953, strain 53_YP3_IM, and four Talas strains.

A characteristic mutation in the modern 2.MED lineage is a G→T substitution at the 613$^{th}$ position in the periplasmic nitrate reductase gene *napA*, introducing a premature TAA stop codon [11,32,33]. This mutation results in a truncated, non-functional protein, explaining the denitrification inability of MED strains. Most of the isolates used in this study exhibit this mutation, which disables nitrate reduction activity. This mutation was not found in the reference *Y. pseudotuberculosis* IP 32953, the strain 53_YP3_IM, the Talas isolates, and six other strains: 31_YP22_SZ, 34_YP28_SZ, 38_YP25_SZ, 26_YP30_SZ, 27_YP31_SZ and 28_YP29_SZ, which were identified by us as belonging to the ANT biovar (S4 Table). Interestingly, strain 57_YP22_IM, which showed similarity to *Y. pseudotuberculosis* IP 32953 in all previous genotype tests, also has a truncated *napA* gene due to the exact same mutation.

All four isolates from the Talas focus had a T-insertion at nucleotide position 303 in the *ssuA* gene, which encodes an ABC transporter protein with the nitrate-binding domain. This insertion, characteristic of the Altaica and Hissarica isolates [34], truncates the protein due to the appearance of a premature stop codon, disabling nitrate reduction in these strains. Another polymorphism of the SsuA protein involves a variable number of KPPSS repeats at the C-terminus of the protein. The majority of this study isolates have three repeats, whereas in *Y. pseudotuberculosis* IP 32953, 53_YP3_IM and 19_YP74_IM, this repeat sequence is present in a single copy. Strain 91_YP6_ADT differs from all other isolates by having two KPPSS repeats at this location. The number of repeats does not appear to affect the denitrification activity of the strains.

Several additional approaches to distinguish between genetic lineages of the main biovars of *Y. pestis* were proposed by Nikiforov et al. (2015) [47]. The authors suggested sets of primers for PCR amplification to identify several marker deletions, without specifying the exact locations of the marker sequences in the genomes. We searched for the primer sequences in the sequenced genomes and found that the 2.ANT/2.MED set targeted an internal part of the carnitine monooxygenase oxygenase subunit *yeaW*, while the Med24 set targeted a non-coding spacer region near the transcriptional stop codon of the 2-octaprenylphenol hydroxylase *ubiL*. It was proposed that the deletion in the 2.ANT/2.MED *yeaW* region was specific to the 2.MED and 2.ANT populations, whereas the Med24 deletion was specific to the main 2.MED1 cluster, distinguishing it from the 2.ANT and.2.MED0 populations.

Sequence comparison revealed that the *yeaW* sequence was conserved in *Y. pseudotuberculosis* and ANT isolates, as well as in strains 19_YP74_IM (Ili intermountain) and 36_YP27_TLH (Talas high mountains), which had previously been identified as similar to MED biovar. In other strains, *yeaW* was highly polymorphic and often truncated, although in some strains, the coding sequences were longer than in the original gene.

The length of the sequence between the Med24 primers (Table 1) varied among different isolates. It measured 233 bp in strain 53_YP3_IM; 231–232 bp in ANT, Talas, and in several MED isolates, including 2_YP67_ADT, 7_YP60_IM, 8_YP55_NP, 12_YP54_IM, 14_YP75_IM, 15_YP61_IM, 24_YP70_IM, 36_YP27_TLH, 57_YP22_IM and 66_YP38_PB. These strains may belong to 2.MED0 population [47]. In all other strains, the sequence was shorter (207 bp) due to a deletion (S4 Table).

Variable Number Tandem Repeat (VNTR) typing and spoligotyping, based on CRISPR-Cas repeats, are popular methods for distinguishing non-main *Y. pestis* biovars [6,7,58,59]. A primer set known as ms06 (AATTTTGCTC-CCCAAATAGCAT and TTTTCCCCATTAGCGAAATAAGTA) targets CRISPR repeat units within a large genomic island (GI) specific to *Y. pseudotuberculosis* and non-main *Y. pestis* isolates [58]. This genomic island is absent in the main biovars of *Y. pestis*. A search for exact matches with the primer sequences identified this CRISPR-Cas repeat element, approximately 1,090 bp in length, along with the corresponding GI in the *Y. pseudotuberculosis* reference strain and in

isolates 19_YP74_IM, 36_YP27_TLH and 57_YP22_IM. Strain 53_YP3_IM contained a homologous GI with the CRISPR element, but mutations were present within the sequence targeted by the direct primer. Homologous GIs with shorter CRISPR repeat elements were also found in *Y. pestis* SCPM reference strains and in the isolate 49_YP5_PAK.

There results of identification of polymorphic loci suggested for *Y. pestis* genotyping in previous publications, are summarized in S3 Table.

**Identification of diagnostic polymorphisms for genotyping of *Y. pestis* isolates belonging to main biovars**

Precise identification of subclades of especially dangerous pathogens is crucial for monitoring the sources and distribution of infectious agents responsible for disease outbreaks. Whole genome sequencing provides comprehensive genetic information about pathogens. However, the precise identification of *Y. pestis* biovars remains challenging.

Fig 2 shows a dendrogram created using progressiveMauve alignment of chromosomal sequences from several selected *Yersinia* isolates and reference genomes. In the dendrogram, the sequences are grouped into two well-separated clusters around the *Y. pseudotuberculosis* and *Y. pestis* reference strains. However, individual strains within these clusters cannot be distinguished from each other due to an insufficient number of genetic variations. To differentiate between the ANT and MED biovars and their biovars, informative sites of genetic polymorphism must be selected.

Identification, validation and verification of diagnostic polymorphic sites suitable for identification of ANT and MED biovars were conducted by utilizing an in-house Python3 script [50]. Workflow of the program is illustrated in S3 Fig.

The program uses archived Illumina *.fastq.gz file copied to input folder for quality control and trimming using Trimmomatic program v0.36 and mapping the reads against the reference sequence with Bowtie2.

The chromosomal sequence of *Y. pseudotuberculosis* IP 32953, representing the putative ancestral state for all *Y. pestis* isolates, was used as the reference. Variant calling was performed using BCFtools. Sequence mismatches, along with variant calling statistics calculated for all isolates, were saved to individual VCF files. Predicted genetic polymorphisms were filtered from indels and additionally by prediction quality parameters using the following thresholds: quality score (QUAL ≥ 30), depth of coverage (DP ≥ 10), mapping quality (MQ ≥ 40), mapping-quality zero fraction (MQ0F ≤ 0.1). The former filtering parameter was used to eliminate polymorphisms unique to individual strains, which may reflect sequencing errors. For each polymorphic locus repeatedly identified at least in 5 individual genomes, 81-bp context sequences

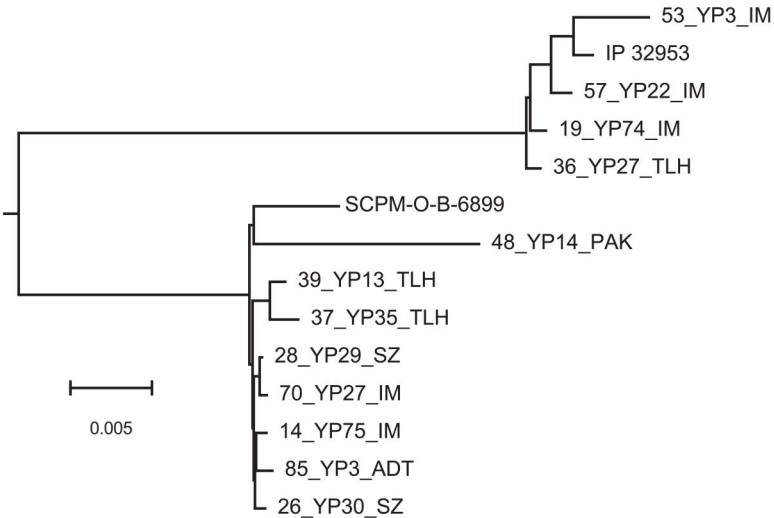

**Fig 2. Phylogenetic tree designed by the progressiveMauve alignment of chromosomal sequences of several selected *Yersinia* isolates and reference genomes.**

surrounding the polymorphic site together with additional metadata regarding the location of the loci within protein-coding genes (CDS) of the reference genome (NC_006155.1) were recorded. The initial dataset comprises 7,007 selected polymorphic loci.

To verify variations in the selected polymorphic loci across the sequenced genomes of *Y. pestis* isolates, local BLASTN alignment was used to identify the locations 81 bp context sequences. BLASTN alignment parameters were set to identify matches of the same length as the queried context sequences, allowing no more than five mismatches (SNPs) per a 81 bp context sequence. This setting ensured the filtering out of false-positive alignments while retaining alignments with a few mismatches, which could result from assembly errors. During this process, the table of polymorphic loci was transformed into a 0/1 matrix, where 0 indicates the successful identification of a homologous context sequence in the assembled genome, and 1 indicates the absence of the reference context sequence in the assembled genome. The resulting matrix was filtered to remove loci where 0 or 1 states occurred in more than 94% of the genomes. The final 0/1 matrix contains 99 polymorphic sites suitable for genotyping (S5 Table). This matrix was used for clustering with the Camin-Sokal parsimony algorithm, which assumes 0s as ancestral states in the genome of *Y. pseudotuberculosis* IP 32953, and 1s as evolutionary progressive states in descendant lineages. Information on the sources of isolation of the *Y. pestis* strains was plotted along the branches of the dendrogram, inferred by the Camin-Sokal parsimony algorithm based on the 0/1 matrix of 99 polymorphic sites, as shown in Fig 3.

It should be noted that the dendrogram shown in Fig 3 should not be interpreted as a phylogenetic tree. The polymorphic sites were specifically selected to distinguish between sub-variants of the dominant MED biovar among the *Y. pestis* isolates. This explains why ANT, Talas, non-main, and *Y. pseudotuberculosis* strains are grouped together in the dendrogram, as the selected polymorphisms are not characteristic of these biovars.

Furthermore, assuming that genetic polymorphisms are driven by selective pressures from adaptation to specific hosts or environmental conditions, it cannot be ruled out that distinct bacteria might develop similar patterns of polymorphic sites through convergent evolution. For example, in a paper by Mas Fiol et al. (2024), it was reported that approximately 36% of genes in sequenced *Y. pestis* genomes harbored convergent (homoplastic) nonsynonymous mutations [60]. This may explain the apparent contradictions in the placement of strains 19_YP74_IM, 57_YP22_IM and 48_YP14_PAK between the trees inferred from chromosomal alignments (Fig 2) and those based on the selected polymorphisms (Fig 3).

As shown in Fig 3, *Y. pestis* isolates that were preliminarily identified as belonging to the MED biovar form two branches in the tree. An attempt was made to identify a possible association between these branches and the sources of isolation of the respective strains. It was found that *Y. pestis* strains in one branch were predominantly isolated from plain desert regions of Central Asia, while the second branch was populated by isolates from upland and mountainous areas, with some level of introgression between the regions, particularly from the uplands toward the surrounding deserts.

The significance of segregation between the desert (n = 37) and upland (n = 47) variants of *Y. pestis* MED genomes was evaluated using the matrix of polymorphic loci (S5 Table) as the feature set. We applied the *RandomForestClassifier* and *model_selection.cross_val_score* functions from the Python *scikit-learn* library (version 1.7.0) in a 5-fold stratified cross-validation scheme, which preserved the original class proportions in each fold. The mean cross-validation accuracy was 0.9889, indicating that the probability of misclassifying isolates between the two environments based on the selected SNPs is about 1%. To further test robustness, we conducted a bootstrap analysis (1,000 permutations) in which clade labels were randomly shuffled, generating a null distribution of accuracies. The graphical output of the analysis is shown in S4 Fig. The observed accuracy exceeded all values from the null distribution, yielding a p-value < 0.001. The same level of confidence, with a p-value of 0.001, was confirmed by the PERMANOVA and Mantel tests. All these tests were implemented in an in-house Python 3 script, available at https://zenodo.org/records/16875681. The matrix of 0/1 states of polymorphic loci (file *example.csv*), formatted for analysis by this program, can be found in the 'input' subfolder after downloading the program from the Zenodo repository.

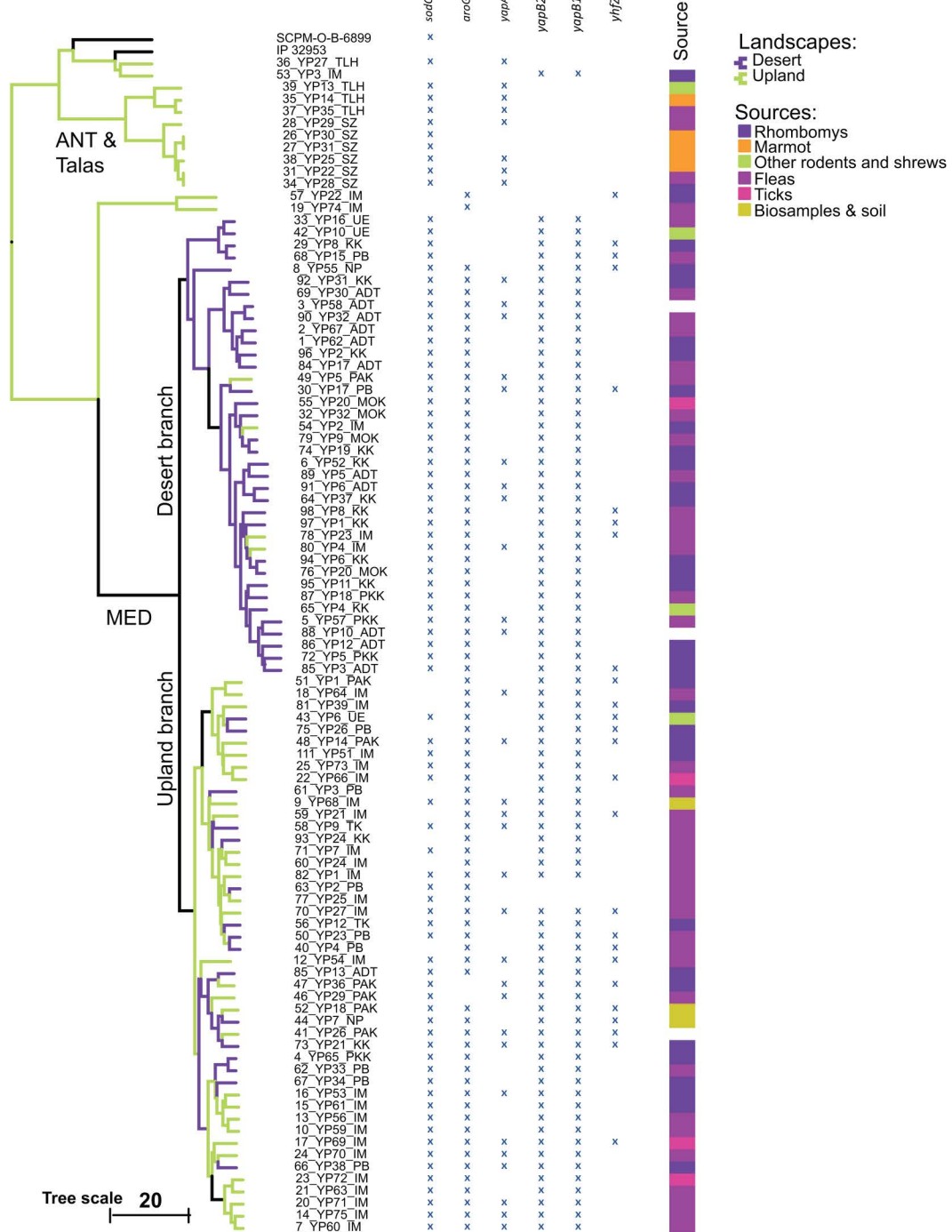

**Fig 3. The dendrogram of clusters of *Yersinia* isolates based on the 0/1 matrix of successful/unsuccessful BLASTN searches for sequences of polymorphic regions across the whole genome sequences of the isolates.** Isolation sources, regions, and landscapes are marked as explained in the figure legend. The results of the BLASTN search for six polymorphic regions within protein-coding genes of the reference strain *Y. pseudotuberculosis* IP 32953 are represented by 'X' symbols, indicating a negative search result, meaning that the region is either missing or modified in the subject genome.

The majority of MED strains of both branches were isolated either from rodent hosts – primarily the great gerbil (*Rhombomys opimus*), though several strains were isolated from the red-tailed gerbil (*Meriones libycus*) and Siberian jerboa (*Allactaga sibirica*), or from fleas (*Xenopsylla gerbilli*, *X. hirtipes*, *X. skrjabini*, *Coptopsylla lamellifer* and *Echidnophaga oschnini*) and ticks (*Hyalommam arginatus*) parasitizing these animals.

The strains of the ANT biovar, the non-main isolates, and those grouped around *Y. pseudotuberculosis* IP 32953 were isolated in mountain valleys. ANT isolates were strictly associated with marmots, particularly the grey marmot (*Marmota baibacina*) and their fleas (*Rhadinopsylla li* subsp. *ventricosa*). Specific *Y. pestis* strains found in the Talas region were isolated from the long-tailed marmot (*M. caudata*), an unidentified shrew (*Sorex* sp.), and unidentified fleas parasitizing marmots. Several upland isolates showed conflicting signals regarding their association with either *Y. pestis* or *Y. pseudotuberculosis*. All were found in mountain valleys. They include strain 36_YP27_TLH, isolated in Talas Mountain region from an unrecorded source, and three strains 53_YP3_IM, 1957_YP22_IM and 19_YP74_IM, isolated in Ili intermountain region from the great gerbil and its flea parasite, *X. gerbilli*. These strains exhibited chromosomal organization similar to *Y. pseudotuberculosis* (Fig 2). However, the allelic states of their polymorphic regions aligned them more closely with the MED biovar (Fig 3). Another isolate with ambiguous taxonomic classification is 48_YP14_PAK, isolated from a great gerbil in Pre-Alakol low mountain region. It has a highly unusual chromosomal structure, distantly resembling those of the Central-Caucasus isolate SCPM (Fig 2), yet it is indistinguishable from the upland branch of the MED biovar based on the states of its polymorphic regions (Fig 3).

Polymorphic sites used to distinguish between the desert and upland branches of the MED biovar (S1 Table) are located predominantly in non-coding regions or hypothetical genes. The allelic states of six polymorphic functional genes across all isolates are shown in Fig 3. These genes are superoxide dismutase (*sodC*), 3-deoxy-7-phosphoheptulonate synthase (*aroG*), pertactin family virulence factors (*yapA*, *yapB1*, and *yapB2*), and a hypothetical gene (*yhfZ*).

In all genomes except for *Y. pseudotuberculosis* IP 32953, the *yhfZ* gene is truncated and accumulates mutations. In strain 48_YP14_PAK, a transposase insertion occurs at this location, along with an additional large hypothetical gene. The *yapB1* and *yapB2* genes are present as an operon only in *Y. pseudotuberculosis* and related strains, such as 19_YP74_IM and 36_YP27_TLH. In all *Y. pestis* strains, these two genes have merged into one hybrid orf due to a deletion covering the 3'-half of *yapB1* and the 5'-half of *yapB2*, with a polymorphic region in between.

The virulence gene *yapA* contains a highly polymorphic region near its 5'-end. Polymorphisms near the stop codon of *aroG* distinguish MED isolates from other clades, including those of ANT biovar. In contrast, mutations in *sodC* occur randomly across all lineages.

Polymorphisms in non-coding regions were more effective for distinguishing between the desert and upland branches of the MED biovar. All upland strains have a large deletion of more than 100 bp downstream of the *yajF*/*mac* fructokinase gene (in the *Y. pseudotuberculosis* IP 32953 genome, *yajF* has the locus tag YPTB0914). Desert MED strains do not have this deletion. Another polymorphism specific to upland MED isolates is located downstream of the inner membrane protein *ylaC* (YPTB2417) and between the *fabV* enoyl-[acyl-carrier-protein] reductase gene (YPTB3950) and the hypothetical gene YPTB3951. Additionally, 27 out of 38 desert isolates have a specific mutation near the stop codon of the hypothetical gene YPTB2020. All these mutations are potentially suitable for designing diagnostic primers to differentiate between the desert and upland branches of the MED biovar.

## Plasmids and mobile genetic elements

To recover plasmid sequences, a Python pipeline shown in S1 Fig was modified to enable the use of multiline reference sequences. The pipeline is freely available [49]. Plasmid sequences, including pMT1, pCD, pPCP, and pCKF from *Y. pestis* SCPM-O-B-6899, and pYV and pYptb3295 from *Y. pseudotuberculosis* IP 32953, were used as reference sequences. The assembled consensus sequences of pCD and pYV were typically similar. In cases where these two alternatives were available, the consensus sequence with greater depth of coverage was selected.

*Y. pestis* isolates typically contain three virulence plasmids: large pMT1 (96,540 – 101,891 bp), the second-largest pCD (70,310 – 70,609 bp) and the small pPCP (9,611 – 9,714 bp). Alignment of Illumina DNA reads against the chromosomal sequences of the isolates revealed a mean sequence depth ranging from 30 to 80 reads per nucleotide. The mean sequence depth for the pMT1 plasmid was 211 (min: 31, max: 468), for pCDit was 281 (min: 52, max: 638), and for pPCP, it was 1,608 (min: 255, max: 4,991). These results suggest that multiple copies of the plasmids were present in each cell of the sequenced cultures.

A novel small cryptic plasmid, pCKF, has been identified among *Y. pestis* isolates from the Central-Caucasus high-altitude autonomous plague focus in Russia [42]. The strain *Y. pestis* SCPM-O-B-6899, used as the reference genome in this study, represents these Central-Caucasus isolates carrying the pCKF plasmid. Sequence assembly of *Yersinia* isolates from Kazakhstan identified one strain, 42_YP10_UE, which contains an identical plasmid. This strain was isolated from a ground squirrel (*Spermophilus pygmaeus*) in the Ural-Embi autonomous plague focus, which is geographically closest to the Central-Caucasus. Genotyping assigned this strain to the desert MED branch (Fig 3), clearly distinguishing it from *Y. pestis* SCPM-O-B-6899. The depth coverage of the pCKF plasmid in strain 42_YP10_UE was comparable to that of pMT1 and pCD, with depths of 226, 196, and 194 reads per base, respectively. However, the depth coverage of pPCP was significantly higher, at 1,057 reads per base.

Only two plasmids, pMT1 and pPCP, were identified in strain 57_YP22_IM. Strain 53_YP3_IM was found to lack any plasmids. An analysis of SPAdes-assembled contigs for this strain did not reveal any contigs that could be associated with known or unknown plasmids.

In our study, we identified multiple insertions of transposable elements in the sequenced genomes, which belong to eight families (S6 Table). The number of insertions varied from 134 to 144 in *Y. pestis* strains from the main and non-main biovars, whereas in *Y. pseudotuberculosis* and related isolates, there were 37–38 IS elements per genome. The most abundant transposases identified were IS100kyp (IS630 family), ISEc39 (IS481 family), IS1541B and IS200 (both belong to the IS5 family). The TnXax1 transposon of the Tn3 family was found only in a single MED isolate (8_YP55_NP). Transposons of the Tn3 family are common in *Xanthomonas*, where they play a major role in the spread of pathogenicity factors [61], but have not been reported before in *Yersinia*.

The most versatile transposons belonged to the IS3 family, which includes the following IS-elements: IS1661, IS1222, IS1400, ISPa31, ISPa74, ISYpe1, ISYpe8, and ISVisp4. We investigated whether the distribution of these transposons and CRISPR-Cas elements across chromosomes could serve as a marker for distinguishing *Yersinia* biovars (Fig 4).

IS1661 transposons were the most frequently found across all isolates compared to other transposons of the IS3 family. Most ANT and MED isolates from both the desert and upland branches exhibit identical distributions of transposon inserts in their chromosomes: eight IS1661 inserts and single inserts of every other IS element, except for ISVisp4, which is highly specific to the Talas *Y. pestis* isolates. Two MED strains, 14_YP75_IM and 18_YP64_IM, are shown in Fig 4A. Strain 18_YP64_IM along with several other MED strains from both branches, 9_YP68_IM, 42_YP10_UE, 84_YP17_ADT, 91_YP6_ADT, 92_YP31_KK and 98_YP8_KK, have only one CRISPR-Cas insertion in its chromosome, whereas other strains in this group have two CRISPR-Cas insertions (Fig 4A). Another distinguishing feature of some strains in this group, such as ANT strains 26_YP30_SZ (shown in Fig 4A) and 28_YP29_SZ, and MED strains 45_YP8_PAK, 82_YP1_IM and 67_YP34_PB, is that, instead of a tandem insertion of ISYpe1-ISYpe8 transposons in the first chromosomal quartile, they possess an ISYpe8-ISYpe8 tandem. In contrast to the other strains in this group, the transposon/CRISPR-Cas distribution pattern in the Talas isolates (35_YP14_TLH, 37_YP35_TLH and 39_YP13_TLH) is quite similar, except for the presence of an ISVisp4 transposase insertion at the end of the second chromosomal quartile, which has not been identified in any other isolate.

The strains grouped around the reference strain *Y. pseudotuberculosis* IP 32953 all exhibit similar patterns of transposon and CRISPR-Cas distribution (Fig 4B), clearly distinguishable from the previous group of ANT, MED, and Talas strains. They differ in the number of CRISPR-Cas inserts, ranging from two in strain 57_YP22_IM to four in 53_YP3_IM.

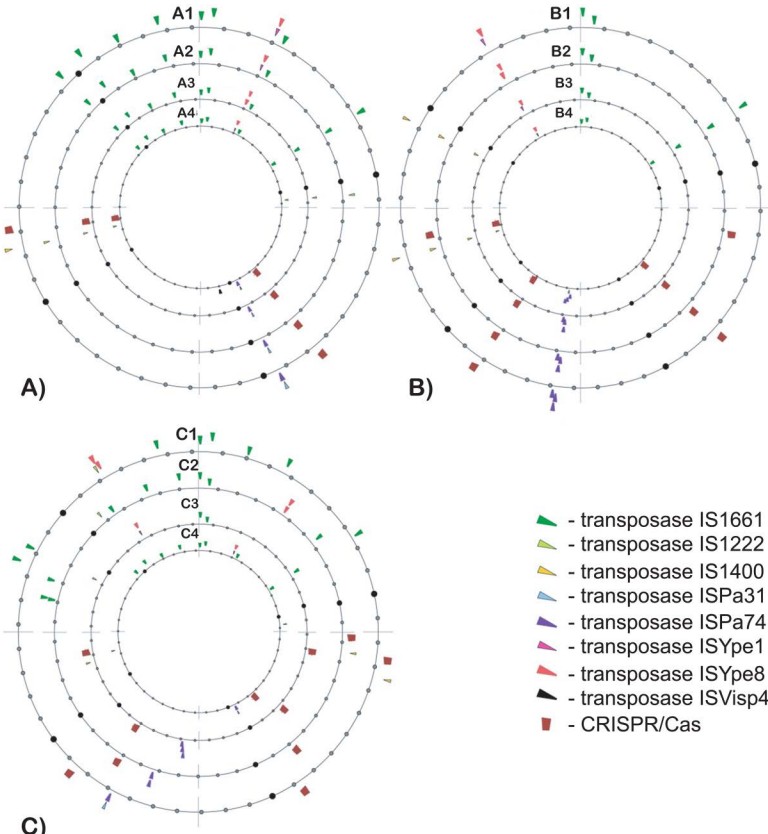

**Fig 4. Atlases of the distribution of transposons of the IS3 family and CRISPR-Cas elements in the genomes of sequenced isolates. A)** Four representative genomes of *Y. pestis* MED, ANT, and Talas biovars: A1 – 14_YP75_IM; A2 – 18_YP64_IM; A3 – 26_YP30_SZ; и A4 – 37_YP35_TLH; **B)** Reference genome of *Y. pseudotuberculosis* IP 32953 and three associated isolates: B1 – IP 32953; B2 – 53_YP3_IM; B3 – 57_YP22_IM; и B4 – 36_ YP27_TLH; **C)** Reference genome of *Y. pestis* SCPM-O-B-6899 and three isolates with unclear taxonomy: C1 – SCPM-O-B-6899; C2 – 48_YP14_PAK, C3 – 19_YP74_IM; и C4 – 42_YP10_UE.

Strain 36_YP27_TLH has four tandem inserts of ISPa74 transposases at the beginning of the third quartile, whereas the other strains contain only three transposases in this tandem. Another distinction was identified in the tandem transposase insertion in the latter half of the fourth quartile: strains 36_YP27_TLH, 57_YP22_IM, and the reference strain *Y. pseudotuberculosis* IP 32953 features ISYpe1-ISYpe8 tandem inserts, while strain 53_YP3_IM contains an ISYpe8-ISYpe8 tandem. Additionally, strain 36_YP27_TLH lacks one IS1400 transposon insertion in the fourth quartile.

Fig 4C illustrates the distribution of transposons and CRISPR-Cas elements in the reference strain *Y. pestis* SCPM-O-B-6899 and three isolates with unclear taxonomy: 48_YP14_PAK, 19_YP74_IM and 42_YP10_UE. The pattern of transposons and CRISPR-Cas elements on the chromosome of strain 19_YP74_IM was identical to that of the *Y. pseudotuberculosis*-associated strain 53_YP3_IM. Both strains contain four CRISPR-Cas inserts, surpassing all other *Yersinia* isolates in this respect. Strain 42_YP10_UE is highlighted here as it is the only strain among *Yersinia* isolates from Kazakhstan to contain the plasmid pCKF. The distribution of transposons on the chromosome of this strain is typical for MED and ANT isolates, with the only difference being the presence of a single CRISPR-Cas insertion. Other MED strains with a single CRISPR-Cas insertion were discussed earlier.

The distributions of transposons and CRISPR-Cas elements in the chromosomes of strain 48_YP14_PAK and the reference strain *Y. pestis* SCPM-O-B-6899 were distinct from each other and from all other *Yersinia* isolates, indicating

their unique lineages. Both strains contain CRISPR-Cas element insertions similar to those found in *Y. pseudotuberculosis* strains, but the sequence repeat blocks within these elements were nearly three times shorter than those in *Y. pseudotuberculosis* strains (see S3 Table).

## Discussion

Globally, *Y. pestis* diversity is structured into several established biovars associated with geographic foci and specific hosts. The separation of the pathogen population into distinct biovars may reflect both the ancient evolutionary history of the species and its adaptation to environmental conditions, as well as to the hosts and insect vectors. The aim of this study was to evaluate the potential of high-throughput sequencing technologies for monitoring the distribution and introgression of *Y. pestis* biovars and subclades associated with different environments and natural plague foci in Central Asia.

Regarding the genetic variability of *Y. pestis* and the related *Y. pseudotuberculosis* species, it is important to note that these organisms are genetically homogeneous [7] even compared to other well-known clonal microorganisms like *Mycobacterium tuberculosis*. While *M. tuberculosis* populations can be characterized by approximately 80,000 polymorphisms [62], with around 10,000 being significant for lineage identification, our variant-calling analysis of *Y. pestis* isolates against the *Y. pseudotuberculosis* reference sequence identified only 7,007 polymorphic sites, many of which were unique to individual strains. Ultimately, only 89 polymorphic sites were sufficiently variable to distinguish between *Y. pestis* isolates. In a separate study by Huang et al. (2023), 1,000 allelic states were proposed for cgMLST typing across eight *Yersinia* species [63]. This suggests that *Y. pestis* is either a much younger pathogenic species compared to *M. tuberculosis* or that its genome is under stronger purifying selection.

Many studies have attempted to define biovars of *Y. pestis* by grouping strains based on identified polymorphisms [6,7,37,59,64,65]. Several biovars have been proposed for main epidemic strains, such as ANT, MED, ORI, and non-main ancient Pestoides strains [11,66–68], which were further subdivided into smaller groups and variants. When a group of strains with a specific polymorphism is identified, researchers often, either explicitly or implicitly, assume that the group is phylogenetically derived from a recent common ancestor, overlooking the possibility that similar mutations could have arisen independently through convergent evolution in response to adaptation to the same hosts or similar environmental conditions.

In this study, we demonstrated the genetic variability of *Yersinia* strains isolated from natural plague foci across the vast desert and mountainous regions of Central Asia (S1 Table). Microbiological characterization of the isolates by phenotype, PCR-based genotyping using standard primer sets, and Illumina whole-genome sequencing were performed to assign the isolates to genetic variants, estimate the level of genetic variability, and explore possible associations between different clades and their areas of isolation or specific mammalian hosts.

*Y. pestis* strains identified as the MED biovar (population 2.MED1) were the most abundant among *Yersinia* isolates from Central Asia and Caucasus. These strains were strongly associated with gerbil populations, their fleas, and ticks, which may play a role in the distribution of the infection [23,69,70]. We developed computational pipelines for the assembly and genotyping of the isolates, based on 99 selected genetic polymorphisms (Figs 2 and 4). This analysis revealed a clear separation of the 2.MED1 strains into two clusters, primarily isolated from desert and upland regions of Central Asia. The segregation significance was supported by the Random Forest classifier (accuracy = 0.9889), PERMANOVA test (F = 21.959), and Mandel test (r = 0.2707).

To estimate the intensity of introgression between subpopulations shown in Fig 3, the isolates were grouped by their sources of isolation from desert and upland plague foci (see Fig 1), irrespective of their genotype. The values of the segregation statistics decreased: classification accuracy = 0.7625, PERMANOVA F = 8.938, Mantel *r* = 0.1244, indicating a substantial level of cross-population introgression. The discordance rate *p* was estimated as the difference between genotype-based and ecotope-based classification accuracies, and this value was converted by Eq. 1 into a 95% CI (16%–34%) for introgression frequency. Introgressions from upland foci into desert foci outnumbered those in the opposite

direction (Fig 3). Among 50 isolates from the desert foci, 16 (32%) carried genotypes characteristic of the upland regions, whereas only 4 of 34 isolates from the upland foci (11.8%) carried the desert-specific genotype. Estimation of the 95% CI using the Wilson score interval (Eq. 1) showed that the frequency of introgressions from upland regions into deserts ranged from 20.8% to 45.8%, whereas the reverse rate ranged from 4.7% to 26.6%.

In general, *Yersinia* strains isolated from uplands and mountain valleys were genetically more diverse than those from the desert plains of Central Asia. This can be attributed to the isolation of host populations in mountainous regions. Over-population of rodents in confined mountain valleys likely drives animals to migrate toward surrounding deserts, creating a flow of migrants from the mountains to the desert regions. This continuous movement may limit the influx of bacterial pathogen variants from external sources [71]. This assumption is supported by the higher observed frequency of intro-gression of upland variants of *Y. pestis* into desert areas, compared to the reverse (Fig 3).

Other genetic lineages of *Yersinia*, such as ANT isolates, and stains from the Talas focus, and several strains showing similarity to *Y. pseudotuberculosis*, were isolated from mountainous regions. Additionally, in contrast to the MED isolates, the ANT and Talas isolates were associated with marmot populations: specifically, with the grey marmot in the case of ANT, and the long-tailed marmot in the case of the Talas isolates (S1 Table and Fig 3) that corroborates with the previous study [72].

An open question remains as to whether the identified groups of *Y. pestis* isolates represent taxonomic units sharing common ancestors or are the result of parallel evolutionary events, where bacterial strains from diverse lineages have converged to similar patterns of allelic states through adaptation to similar conditions. Our analysis of whole-genome sequences suggests that these groups of isolates are more likely polyphyletic.

Comparison of whole-genome sequences (Fig 2) demonstrated a clear separation between *Y. pestis* strains and *Y. pseudotuberculosis*. Within the *Y. pestis*, ANT and MED strains clustered together forming mixed sub-clusters indicating their close relatedness to each other within the main biovar.

The most striking evidence of convergent evolution was observed in strains 57_YP22_IM, 19_YP74_IM and 48_YP14_PAK. The first two strains, isolated from great gerbils and their fleas in the Ili intermountain autonomous plague focus, share common genome organization with *Y. pseudotuberculosis* reference strain IP 32953 (Fig 2). Their close relationship with this species is also evident from the distribution of transposons and CRISPR-Cas elements (Fig 4). However, bacteriological tests identified these strains as *Y. pestis*, and their genetic polymorphisms grouped them between the ANT and MED clusters (Fig 3). Strain 48_YP14_PAK, isolated from a great gerbil in Pre-Alakol low mountain plague focus, belongs to a distinct *Y. pestis* lineage, showing no similarity to any other isolates (Figs 2 and 4). Nevertheless, it exhibited a pattern of polymorphic alleles indistinguishable from those of other *Y. pestis* upland isolates (Fig 3). These findings led us to conclude that adaptation to mammalian hosts and environmental conditions played a much more significant role in shaping the pattern of polymorphic alleles in the *Yersinia* population than shared ancestry. In a recent publication, this evolutionary scenario was suggested for *Y. pestis* isolates from Africa [60].

In this study, we were unable to identify polymorphisms with strong, unambiguous phylogenetic signals, with the possible exception of nitrate reduction dysfunction associated with a deletion in the *napA* gene, which is characteristic of all MED strains (S3 Table). The absence of deletions in *ilvN*, the deletion in *araC*, and the G allelic state at the 671$^{st}$ position of *rhaS* identified Talas strains as non-main biovar isolates that is aligned with a previous publication [56,57]. However, all other tests indicated a close relationship with ANT isolates. This may result from the adaptation of both lineages to similar hosts – grey and long-tailed marmots [71].

Several diagnostic primers have been proposed to distinguish subgroups of the MED and ANT biovars [47], particularly targeting the Med24 deletion in the spacer region between the *yeaW* and *ubiI* genes, a feature typical of MED strains but absent in 2.MED0 group. We identified several MED strains from both desert and upland branches that lacked this deletion. Mutations near the stop codon of the monooxygenase subunit *yeaW* were present in all MED isolates, including the non-MED strain 48_YP14_PAK. However, alignment of these genes suggested that these mutations arose from multiple parallel events rather than a single ancestral mutation.

A notable feature of 2.MED0 strains isolated from the Caucasus and Caspian Sea regions was the presence of a novel cryptic plasmid, pCKF [17]. We found only one strain, 42_YP10_UE, containing this plasmid. This strain was isolated in Ural-Embi desert near Caspian Sea. Interestingly, this strain carried the Med24 deletion typical of most MED isolates from Central Asia and showed no similarity with the reference strain *Y. pestis* SCPM-O-B-6899 from Central-Caucasus, which also carried the pCKF plasmid. This finding suggests that the pCKF plasmid is not restricted to a specific *Y. pestis* clade and can spread across lineages. This information is concerning. Although there is no evidence linking this plasmid to increased virulence, its spread to other regions indicates that it confers some advantage to the pathogen. The plasmid contains eight genes, including *virB5* and *virB6*, which encode virulence-associated type 4 secretion system proteins [42,17].

The analysis of the distribution of different types of transposons and CRISPR-Cas elements on the chromosomes demonstrated the robustness of the method in identifying different lineages of *Y. pestis* and *Y. pseudotuberculosis* (Fig 4). Although the method is applicable for identifying distinct *Yersinia* biovars and subclades [35,73], it should not be used for phylogenetic inferences, as no computational models currently exist to translate differences in transposon distribution patterns into evolutionary distances. The comparison of transposon and CRISPR-Cas distribution patterns aligned with the results of chromosome alignment (Fig 2), suggesting that the main ANT and MED biovars can be polyphyletic groups formed through the convergence of several *Y. pestis* lineages, driven by adaptation to the same mammalian hosts. In Central Asia, these hosts include desert and upland gerbil populations for the MED biovar, and marmot populations for the ANT biovar. Environmental conditions also play a role in shaping specific genotypes, which could explain the division of MED *Y. pestis* into desert and upland branches.

## Conclusion

The major discovery of this study is the demonstration of genetic diversity among the dominant MED *Y. pestis* isolates in Central Asia, revealing their division into desert and upland subclades. Within the broader framework, the desert and upland variants described here represent ecologically differentiated subclades of the same globally recognized MED biovar. The genomic differences identified between these subclades contribute to a more nuanced understanding of *Y. pestis* population structure and its ecological adaptation.

However, it was shown that these groups are polyphyletic in terms of phylogeny, as adaptation to different mammalian hosts and environmental conditions has a greater influence than ancestral relationships between *Y. pestis* and *Y. pseudotuberculosis* isolates. While the identified polymorphisms may help determine the origin of an isolate, they are not suitable for phylogenetic inferences. Researchers working with these pathogens should be cautious when inferring relationships between isolates based on a simple count of shared or differing SNPs and indels.

The panel of *Y. pestis* genetic polymorphisms developed in this study provides a valuable tool for high-resolution surveillance of pathogens, particularly for distinguishing MED biovar strains into ecologically meaningful upland and desert branches. These polymorphisms, combined with the analysis of mobile genetic elements such as transposons, CRISPR-Cas systems, and plasmids including the concerning detection of the pCKF plasmid in a non-related isolate, can aid in tracking lineage distribution, potential host adaptation, and horizontal gene transfer across plague foci. Integration of these approaches into routine plague monitoring can enable rapid and accurate lineage identification, facilitating timely source attribution during outbreak investigations. This enhanced resolution supports more precise risk assessments in endemic areas by identifying ecologically specialized variants with differing potentials for host range expansion or virulence. The approach demonstrated here, grounded in the combination of whole-genome sequencing with robust statistical validation, can be readily incorporated into national and international plague surveillance frameworks, thereby strengthening preparedness and early-warning capabilities for the emergence or re-emergence of *Y. pestis* variants of public health concern.

A concerning finding of this study was the discovery of the potentially virulence plasmid pCKF, which was first identified in the Central-Caucasus region, in a genetically unrelated *Y. pestis* isolate. This may indicate that the plasmid has spread

from its original point of acquisition within the *Yersinia* population to other regions. Additionally, the insertion of the TnXax1 transposon in one of the sequenced *Y. pestis* genomes suggests genetic material exchange with other pathogens. These developments illustrate the ongoing evolution of the pathogen and warrant close monitoring and control.

While this study focuses specifically on *Y. pestis* isolates from Central Asia, the genomic patterns observed, particularly those related to ecological adaptation and genetic variations, highlight the potential value of integrating regional datasets into broader surveillance frameworks. Future studies should include representative strains from other geographic regions to assess the universality of the identified polymorphisms. Such comparative analyses would be essential for aligning regional genomic surveillance efforts with global strategies for monitoring *Y. pestis* evolution and outbreak potential.

## Supporting information

**S1 Fig.  Pipeline of assembly and annotation of genomes of *Yersinia* isolates.**
(PDF)

**S2 Fig.  Shapes of colonies of *Y. pestis* exemplified by strain 9_YP68_IM: A) after 12 hours; B) after 24 hours; and C) after 48 hours on Hottinger agar at 28°C; and D) after 24 hours on Hottinger agar supplemented with 1% blood hemolysate.**
(TIF)

**S3 Fig.  Pipeline of identification of polymorphic genomic regions suitable for genotyping.**
(PDF)

**S4 Fig.  Bootstrap null distribution of classification accuracy across the desert (37 strains) and upland (47 strains) variants of *Yersinia pestis* isolates.** Histogram shows the distribution of classification accuracies obtained from 1,000 Random Forest runs with shuffled group labels (null distribution). The red dashed vertical line indicates the actual classification accuracy 0.9889 achieved using true group labels. The area of the histogram to the right of the left score represents the empirical accuracy classification value, estimating the probability of observing such classification performance by chance. This analysis supports the statistical significance of group separability based on the polymorphic features in the input matrix.
(PDF)

**S1 Table.  *Yersinia pestis* strains used in this study.**
(XLSX)

**S2 Table.  Phage susceptibility, biochemical activities and results of diagnostic PCR amplification obtained for selected *Yersinia* isolates.**
(DOCX)

**S3 Table.  Genome assembly statistics.**
(XLSX)

**S4 Table.  Results of computer-based genotyping of the isolates by allelic states of classical marker genes and VNTR microsatellite spots.**
(DOCX)

**S5 Table.  Matrix of polymorphic genetic loci.**
(XLSX)

**S6 Table.  Numbers of transposons and CRISPR-Cas elements identified in different *Yersinia* strains.**
(DOCX)

## Acknowledgments

All authors of the paper tank Prof. Alim Masgutovich Aikimbayev for his support, reviewing and recommendations regarding the manuscript and plague pathogen biology.

## Financial disclosure

This work was supported by the Minister of Science and Higher Education of the Republic of Kazakhstan; Grant AP19680079, "Study of molecular genetic features and variability of plague and tularemia strains in epidemiological surveillance of zoonoses" to AAA. The funder had no role in study design, data collection and analysis, decision to publish, or preparation of the manuscript.

## Author contributions

**Conceptualization:** Aigul A. Abdirassilova, Zauresh B. Zhumadilova, Vladimir L. Motin, Oleg N. Reva.

**Data curation:** Aigul A. Abdirassilova, Zauresh B. Zhumadilova, Gulnara Zh. Tokmurziyeva, Ziyat Zh. Abdel, Vladimir L. Motin, Oleg N. Reva.

**Formal analysis:** Aigul A. Abdirassilova, Altynai K. Kassenova, Beck Z. Abdeliyev, Saule K. Umarova, Oleg N. Reva, Altyn K. Rysbekova.

**Funding acquisition:** Aigul A. Abdirassilova.

**Investigation:** Aigul A. Abdirassilova, Duman T. Yessimseit, Altynai K. Kassenova, Beck Z. Abdeliyev, Ziyat Zh. Abdel, Tatiyana V. Meka-Mechenko, Elmira Zh. Begimbayeva, Sanzhar D. Agzam, Oleg N. Reva, Altyn K. Rysbekova.

**Methodology:** Aigul A. Abdirassilova, Duman T. Yessimseit, Ziyat Zh. Abdel, Tatiyana V. Meka-Mechenko, Elmira Zh. Begimbayeva, Sanzhar D. Agzam, Oleg N. Reva, Altyn K. Rysbekova.

**Project administration:** Aigul A. Abdirassilova.

**Resources:** Aigul A. Abdirassilova.

**Software:** Oleg N. Reva.

**Supervision:** Aigul A. Abdirassilova, Zauresh B. Zhumadilova.

**Validation:** Aigul A. Abdirassilova, Oleg N. Reva.

**Visualization:** Aigul A. Abdirassilova, Oleg N. Reva.

**Writing – original draft:** Aigul A. Abdirassilova, Duman T. Yessimseit, Altynai K. Kassenova, Beck Z. Abdeliyev, Gulnara Zh. Tokmurziyeva, Galina G. Kovaleva, Saule K. Umarova, Vladimir L. Motin, Oleg N. Reva, Altyn K. Rysbekova.

**Writing – review & editing:** Aigul A. Abdirassilova, Duman T. Yessimseit, Vladimir L. Motin, Oleg N. Reva, Altyn K. Rysbekova.

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
