## [Decision Letter · Decision Letter 0]

22 Jun 2025

Whole genome sequencing of Yersinia pestis isolates from Central Asian natural plague foci revealed the role of adaptation to different hosts and environmental conditions in shaping specific genotypes

Dear Dr. Reva,

Thank you for submitting your manuscript to PLOS Neglected Tropical Diseases. After careful consideration, we feel that it has merit but does not fully meet PLOS Neglected Tropical Diseases's publication criteria as it currently stands. Therefore, we invite you to submit a revised version of the manuscript that addresses the points raised during the review process.

Please submit your revised manuscript within 60 days Aug 21 2025 11:59PM. If you will need more time than this to complete your revisions, please reply to this message or contact the journal office at plosntds@plos.org. Please include the following items when submitting your revised manuscript:

We look forward to receiving your revised manuscript.

Kind regards,

Ben Pascoe

Academic Editor

Mathieu Picardeau

Section Editor

Shaden Kamhawi

co-Editor-in-Chief

Paul Brindley

co-Editor-in-Chief

**Additional Editor Comments:**

After careful consideration of two expert reviews, we believe the manuscript can reach publishable quality, but substantial revision is required.

In brief:

• Please articulate an explicit, testable hypothesis and streamline the Introduction to highlight the specific knowledge gap addressed.

• Descriptive SNP/parsimony analyses alone are insufficient. Add a statistically supported phylogeny and at least one population-structure or phylogeographic test linking genotype to ecology/host.

• Explain how strains were chosen from the national collection and acknowledge the inherent sampling imbalance when interpreting ecological patterns.

• Provide an explicit statement of BSL-3 approvals and regulatory compliance for work with Y. pestis.

• Improve figure resolution (vector graphics), split intermixed Results/Discussion text, and enforce consistent terminology for biovars, clades, and strain names.

• Correct Table 1 primer information; supply amplicon sizes, biovar specificity, and software versions/parameters for all tools.

• Add a concise paragraph on how your SNP panel or plasmid findings could aid surveillance and explicitly list analytical limitations (e.g., lack of long-read data, small subgroup sizes).

**Journal Requirements:**

1) Please provide an Author Summary. This should appear in your manuscript between the Abstract (if applicable) and the Introduction, and should be 150-200 words long. The aim should be to make your findings accessible to a wide audience that includes both scientists and non-scientists. Sample summaries can be found on our website under Submission Guidelines:

3) Some material included in your submission may be copyrighted. According to PLOSu2019s copyright policy, authors who use figures or other material (e.g., graphics, clipart, maps) from another author or copyright holder must demonstrate or obtain permission to publish this material under the Creative Commons Attribution 4.0 International (CC BY 4.0) License used by PLOS journals. Please closely review the details of PLOSu2019s copyright requirements here: PLOS Licenses and Copyright. If you need to request permissions from a copyright holder, you may use PLOS's Copyright Content Permission form.

Potential Copyright Issues:

i) Please confirm (a) that you are the photographer of S2, or (b) provide written permission from the photographer to publish the photo(s) under our CC BY 4.0 license.

ii) Figure 1. Please (a) provide a direct link to the base layer of the map (i.e., the country or region border shape) and ensure this is also included in the figure legend; and (b) provide a link to the terms of use / license information for the base layer image or shapefile. We cannot publish proprietary or copyrighted maps (e.g. Google Maps, Mapquest) and the terms of use for your map base layer must be compatible with our CC BY 4.0 license.

4) Please amend your detailed Financial Disclosure statement. This is published with the article. It must therefore be completed in full sentences and contain the exact wording you wish to be published.

**Reviewers' Comments:**

Reviewer's Responses to Questions

**Key Review Criteria Required for Acceptance?**

**Methods**

-Are the objectives of the study clearly articulated with a clear testable hypothesis stated?

-Is the study design appropriate to address the stated objectives?

-Is the population clearly described and appropriate for the hypothesis being tested?

-Is the sample size sufficient to ensure adequate power to address the hypothesis being tested?

-Were correct statistical analysis used to support conclusions?

-Are there concerns about ethical or regulatory requirements being met?

Reviewer #1: (No Response)

Reviewer #2: Are the objectives of the study clearly articulated with a clear, testable hypothesis stated?

Partially.

The manuscript has well-described aims: to characterise Y. pestis isolates from Central Asia and identify genetic polymorphisms associated with ecological adaptation and lineage divergence. However, a clear hypothesis is not explicitly stated, and the narrative blends surveillance goals with exploratory phylogenomic analysis. For a stronger framing, the authors should articulate a central hypothesis (e.g., “Y. pestis genomic variation correlates with host or environmental origin more than with clonal ancestry”) early in the Introduction.

Is the study design appropriate to address the stated objectives?

Yes, broadly.

The use of WGS for 98 Y. pestis isolates, with phenotypic testing and in silico variant analysis, is appropriate for evaluating biovar classification, polymorphisms, and lineage diversity. However, comparative analyses with global strains are missing, which limits broader inference and validation of their findings.

Is the population clearly described and appropriate for the hypothesis being tested?

Yes.

The manuscript provides detailed ecological and geographic context for the 98 isolates, spanning desert and upland plague foci. The inclusion of multiple hosts (e.g., gerbils, marmots, fleas) is appropriate given the zoonotic ecology of plague. A more structured breakdown of how many isolates came from which host species or region would help clarify population stratification.

Is the sample size sufficient to ensure adequate power to address the hypothesis being tested?

Generally yes, but with caveats.

For regional genomic surveillance, the sample size is robust (n = 98). However, comparisons across clades or ecological categories (e.g., desert vs upland) sometimes involve much smaller groups (e.g., 4 Talas isolates), which limits statistical power. The authors should acknowledge this limitation, particularly when making ecological inferences.

Were correct statistical analyses used to support conclusions?

Partially.

The authors use custom SNP calling, clustering, and dendrogram construction, but they rely heavily on parsimony-based clustering and descriptive comparisons without formal phylogenetic or population structure modelling. No statistical tests (e.g., Mantel tests, AMOVA, PCA) are used to quantify associations between genotype and geography/host. This weakens the analytical rigour and should be addressed with additional comparative or phylogenetic methods if claims about convergent evolution or ecological adaptation are to be substantiated.

Are there concerns about ethical or regulatory requirements being met?

No major concerns raised, but additional clarity would help.

The authors state that isolates came from a national strain collection and were collected during routine surveillance. However: There is no explicit statement of ethical or biosafety approvals for working with Y. pestis (a BSL-3 agent).

**Results**

-Does the analysis presented match the analysis plan?

-Are the results clearly and completely presented?

-Are the figures (Tables, Images) of sufficient quality for clarity?

Reviewer #1: (No Response)

Reviewer #2: Does the analysis presented match the analysis plan?

Broadly yes, but exploratory in nature.

The study presents genome sequencing, variant detection, plasmid profiling, and phenotypic tests of Y. pestis isolates as planned. The described pipelines and laboratory methods align with the Results section. However, because the manuscript is exploratory rather than hypothesis-driven, there is no formal analysis plan or pre-specified comparisons. The selection of SNPs and construction of polymorphism matrices are well described, but the lack of formal statistical testing makes the structure more observational than analytical.

Are the results clearly and completely presented?

Partially.

The authors provide extensive detail, including strain-level variation, allele states, plasmid content, and CRISPR/transposon profiles. However, the narrative is dense, with long blocks of text that lack synthesis or summarisation.

Some key findings (e.g., upland vs desert divergence, pCKF identification) are buried in technical detail, making them hard to extract.

The lack of statistical tests or quantitative summaries (e.g., number of unique SNPs per clade, plasmid prevalence by ecotype) weakens the clarity of the results.

Recommendation: Add a concise summary table per major section (e.g., SNP patterns, plasmid presence, phenotypic test outcomes) and integrate clearer narrative transitions.

Are the figures (Tables, Images) of sufficient quality for clarity?

No

Several figures are blurry or low-resolution, particularly Figures 2–5, which are central to the interpretation of genomic variation and lineage structure.

Font sizes are too small in dendrogram labels and SNP context annotations.

Figure 3 (pipeline diagram) is unnecessary and should be removed or moved to Supplementary Material.

Supplementary tables are useful but need better integration into the main narrative.

**Conclusions**

-Are the conclusions supported by the data presented?

-Are the limitations of analysis clearly described?

-Do the authors discuss how these data can be helpful to advance our understanding of the topic under study?

-Is public health relevance addressed?

Reviewer #1: (No Response)

Reviewer #2: Are the conclusions supported by the data presented?

Partially.

The conclusions - particularly regarding the ecological structuring of Y. pestis genotypes (upland vs desert) and the possibility of convergent evolution - are intriguing and plausibly supported by the data. However, some conclusions (e.g., strong claims about adaptive convergence or plasmid transfer across lineages) go beyond the data, given the limitations of short-read sequencing and lack of phylogenetic inference.

The authors also understate the ambiguity around taxonomic assignment of outlier strains (e.g., Y. pseudotuberculosis-like genomes classified as Y. pestis).

Are the limitations of the analysis clearly described?

The authors do not sufficiently acknowledge several key limitations:

No long-read data for plasmid resolution or structural variation.

Small subgroup sizes for comparisons (e.g., Talas isolates).

Reliance on selected SNPs and parsimony trees rather than robust phylogenetic or population genetic approaches.

No formal statistical or model-based testing of associations between genotype, host, or environment.

Recommendation: A dedicated paragraph in the Discussion should explicitly list these analytical limitations and their implications for interpretation.

Do the authors discuss how these data can be helpful to advance our understanding of the topic under study?

Yes, in part.

The study contributes a valuable genomic dataset from a poorly sampled geographic region. The authors suggest that their polymorphism panel could be used for biovar identification and ecological surveillance. However, they could better articulate:

How these tools might integrate with global Y. pestis surveillance frameworks, or

How their SNPs or plasmid findings compare to those in outbreak strains from other continents.

Is public health relevance addressed?

Minimally.

The manuscript briefly mentions zoonotic risk, ecological change, and the importance of surveillance in Kazakhstan, but does not connect this meaningfully to:

The risk of human outbreaks, or

How these findings might guide vector control, diagnostics, or cross-border health policy.

Recommendation: Strengthen the final paragraph of the Discussion to explicitly state how the findings could improve public health risk assessment, surveillance tools, or outbreak preparedness.

**Editorial and Data Presentation Modifications?**

Reviewer #1: (No Response)

Reviewer #2: Clarify and streamline the Introduction

While informative, the Introduction could be shortened and better focused. The authors should clearly articulate the gap this study addresses and explicitly state a testable hypothesis (e.g., genomic variation in Y. pestis correlates with ecological origin more than with shared ancestry).

Improve logical flow and paragraphing

The Results and Discussion sections are dense and occasionally read like internal notes. Use clearer topic sentences and summary statements to guide the reader through complex genomic findings.

Correct typographical and language issues

Line 29: "sequnced" > "sequenced"

Line 33: "suitable tool" > "suitable tools"

Line 54: "ware caused" > "were caused"

Consider a light language edit throughout to improve fluency and clarity.

Ensure consistency in terminology

Terms like “biovar,” “clade,” “subvariant,” and “non-main” are used inconsistently. Define and standardise these terms early, especially when linking phenotypic biovars to genomic groupings.

Add ethical and biosafety statement

Given that Y. pestis is a BSL-3 pathogen, include a statement affirming compliance with appropriate ethical and biosafety regulations, including approvals for handling high-risk agents.

Improve figure resolution and legibility

Figures 2–5 are critical to the paper’s conclusions but are currently low-resolution with small or pixelated text. Please:

Re-export or redraw figures using vector formats (PDF, SVG) to ensure clarity.

Increase font sizes for isolate names and branch labels.

Ensure all abbreviations, symbols, and colour keys are defined in the legends.

Remove or relocate Figure 3

The diagram showing the internal SNP-calling pipeline (Figure 3) is not informative to the scientific conclusions and can be removed or moved to supplementary materials if needed for pipeline documentation.

Improve integration of supplementary tables

The manuscript frequently references Suppl. Tables (e.g., S1–S5), but their use would be clearer if specific examples were summarised in the main text, and/or a “key findings” summary table were added (e.g., SNP markers per lineage, plasmid carriage, or phenotypic traits by clade).

Explicitly list software versions

To ensure full reproducibility, please add version numbers for all software tools used, including:

Trimmomatic

SPAdes

Bowtie2

BCFtools

Prokka

RagTag

BLASTN

Python version for custom scripts

Even if run with default parameters, this should be stated explicitly.

Discuss limitations of short-read data

The inability to resolve plasmid structure, including the circularisation or structural variation of pCKF and other elements, should be clearly stated. The authors should note that long-read sequencing would provide more definitive plasmid resolution.

Add global context

The manuscript would be strengthened by discussing how these Central Asian lineages relate to global Y. pestis diversity, including pandemic-associated clades or modern outbreak strains. A phylogenetic comparison or even literature summary would enhance impact.

Strengthen public health relevance

Currently, the manuscript emphasises ecological and genetic findings but says little about how these data could inform surveillance, risk assessment, or control. Consider adding a concluding paragraph that:

Links the findings to public health surveillance,

Highlights diagnostic implications (e.g., lineage-specific markers),

Discusses the relevance of cryptic plasmids (e.g., pCKF) for virulence monitoring.

**Summary and General Comments**

Reviewer #1: (No Response)

Reviewer #2: The manuscript offers a rare and detailed look at Y. pestis evolution in Central Asia - a region of historical and current epidemiological importance. The work has the potential to substantially contribute to our understanding of plague ecology and evolution, particularly in the context of environmental adaptation and mobile genetic elements. However, its broader impact is currently limited by presentation, analytical, and contextual shortcomings that should be addressed in revision.

PLOS authors have the option to publish the peer review history of their article (what does this mean? ). If published, this will include your full peer review and any attached files.

**Do you want your identity to be public for this peer review?** For information about this choice, including consent withdrawal, please see our Privacy Policy .

Reviewer #1: No

Reviewer #2: No

**Figure resubmission:**

**Reproducibility:**



---

## [Decision Letter · Decision Letter 1]

13 Aug 2025

Response to Reviewers
Revised Manuscript with Track Changes
Manuscript

Kind regards,

Shaden Kamhawi

co-Editor-in-Chief

Paul Brindley

co-Editor-in-Chief

**Additional Editor Comments:**
**Journal Requirements:**

**Reviewers' comments:**

**Key Review Criteria Required for Acceptance?**

**Methods**

-Are the objectives of the study clearly articulated with a clear testable hypothesis stated?

-Is the study design appropriate to address the stated objectives?

-Is the population clearly described and appropriate for the hypothesis being tested?

-Is the sample size sufficient to ensure adequate power to address the hypothesis being tested?

-Were correct statistical analysis used to support conclusions?

-Are there concerns about ethical or regulatory requirements being met?

Reviewer #1: Yes.

Reviewer #2: (No Response)

**Results**

-Does the analysis presented match the analysis plan?

-Are the results clearly and completely presented?

-Are the figures (Tables, Images) of sufficient quality for clarity?

Reviewer #1: Yes.

Reviewer #2: (No Response)

**Conclusions**

-Are the conclusions supported by the data presented?

-Are the limitations of analysis clearly described?

-Do the authors discuss how these data can be helpful to advance our understanding of the topic under study?

-Is public health relevance addressed?

Reviewer #1: Yes.

Reviewer #2: (No Response)

**Editorial and Data Presentation Modifications?**

Reviewer #1: Accept.

Reviewer #2: (No Response)

**Summary and General Comments**

Reviewer #1: The reviewer acknowledges and accepts all responses provided by the authors to the previous comments raised during the review process. No further remarks.

Reviewer #2: This manuscript analyses ~100 Yersinia pestis isolates from Central Asian plague foci using short-read WGS, targeted polymorphism panels, plasmid profiling and classical phenotyping. The dataset is valuable and the revision improves framing, biosafety statements, figure exports, and data availability. However, several core analytical and reporting issues remain.

Major points

Phylogenomic backbone: Provide a core-genome SNP maximum-likelihood tree (with support) for all isolates and use it to anchor ecological interpretations; the current diagnostic-SNP dendrogram is explicitly not a phylogeny.

Genome QC: Add a per-isolate table (genome size, N50/L50, contig count, %GC, mean depth) and state any exclusions.

Statistics: The random-forest classification of desert vs upland needs clearer methods (feature set, CV scheme, class balance) and proper p-value reporting (e.g., p < 0.001, not 0.0). Consider a simple AMOVA/Mantel/PERMANOVA to quantify genotype–ecology association.

Reproducibility: List exact versions/parameters for Trimmomatic, SPAdes, Bowtie2, BCFtools, Prokka, RagTag (and seeds for ML).

Plasmid claims: The pCKF signal is interesting; include read-level evidence in Supplement (coverage plots; discordant pairs). If feasible, note any targeted validation.

Minor points

Tighten language and standardise terminology (biovar/clade/subvariant).

Keep the pipeline graphic in Supplement; ensure all main figures have legible fonts/keys.

Add a brief global context paragraph situating these lineages relative to established Y. pestis diversity.

End with a clearer public-health relevance statement (surveillance utility of markers, implications for risk assessment).

Ethics & data

BSL-3/biosafety statements appear adequate. Data availability is strong; please also provide a pinned, isolate-level metadata CSV.

PLOS authors have the option to publish the peer review history of their article (what does this mean? ). If published, this will include your full peer review and any attached files.

**Do you want your identity to be public for this peer review?** For information about this choice, including consent withdrawal, please see our Privacy Policy .

Reviewer #1: No

Reviewer #2: No

**Figure resubmission:**

**Reproducibility:** To enhance the reproducibility of your results, we recommend that authors of applicable studies deposit laboratory protocols in protocols.io, where a protocol can be assigned its own identifier (DOI) such that it can be cited independently in the future. Additionally, PLOS ONE offers an option to publish peer-reviewed clinical study protocols. Read more information on sharing protocols at https://plos.org/protocols?utm_medium=editorial-email&utm_source=authorletters&utm_campaign=protocols

---

## [Editor Report · Decision Letter 2]

5 Sep 2025

Dear Prof. Reva,

We are pleased to inform you that your manuscript 'Whole genome sequencing of Yersinia pestis isolates from Central Asian natural plague foci revealed the role of adaptation to different hosts and environmental conditions in shaping specific genotypes' has been provisionally accepted for publication in PLOS Neglected Tropical Diseases.

Best regards,

Ben Pascoe

Academic Editor

Mathieu Picardeau

Section Editor

Shaden Kamhawi

co-Editor-in-Chief

Paul Brindley

co-Editor-in-Chief

Thanks for engaging with the review process and addressing all reviewer comments.

---

## [Editor Report · Acceptance letter]

Dear Prof. Reva,

We are delighted to inform you that your manuscript, " 

Whole genome sequencing of Yersinia pestis isolates from Central Asian natural plague foci revealed the role of adaptation to different hosts and environmental conditions in shaping specific genotypes," has been formally accepted for publication in PLOS Neglected Tropical Diseases.

Best regards,

Shaden Kamhawi

co-Editor-in-Chief

Paul Brindley

co-Editor-in-Chief
